# Absolute quantum yield measurements of fluorescent proteins using a plasmonic nanocavity

Daja Ruhlandt[1], Martin Andresen[2], Nickels Jensen[2], Ingo Gregor [1], Stefan Jakobs [2,3,4], Jörg Enderlein [1,4✉] & Alexey I. Chizhik [1✉]

One of the key photophysical properties of fluorescent proteins that is most difficult to measure is the quantum yield. It describes how efficiently a fluorophore converts absorbed light into fluorescence. Its measurement using conventional methods become particularly problematic when it is unknown how many of the proposedly fluorescent molecules of a sample are indeed fluorescent (for example due to incomplete maturation, or the presence of photophysical dark states). Here, we use a plasmonic nanocavity-based method to measure absolute quantum yield values of commonly used fluorescent proteins. The method is calibration-free, does not require knowledge about maturation or potential dark states, and works on minute amounts of sample. The insensitivity of the nanocavity-based method to the presence of non-luminescent species allowed us to measure precisely the quantum yield of photo-switchable proteins in their on-state and to analyze the origin of the residual fluorescence of protein ensembles switched to the dark state.

[1] Georg-August-University Göttingen, Third Institute of Physics - Biophysics, Göttingen, Germany. [2] Department of NanoBiophotonics, Max Planck Institute for Biophysical Chemistry, Am Fassberg 11, 37077 Göttingen, Germany. [3] University of Göttingen Medical Faculty, Clinic of Neurology, Robert-Koch-Strasse 40, 37075 Göttingen, Germany. [4] Cluster of Excellence "Multiscale Bioimaging: from Molecular Machines to Networks of Excitable Cells," (MBExC), University of Göttingen, Göttingen, Germany. ✉email: jenderl@gwdg.de; alexey.chizhik@phys.uni-goettingen.de

Fluorescent proteins (FPs) have become one of the most important tools of bio-imaging and super-resolution microscopy[1–3]. They allow for observing biological structures and dynamics in live cells, tissues, and whole organisms. Special variants of these proteins (photoswitchable proteins), which exhibit reversible transitions between an emissive on-state and non-emissive off-states, have found broad application in sub-diffraction super-resolution microscopy, allowing to resolve structural details much beyond the diffraction limit of resolution in conventional fluorescence microscopy[4]. However, using these proteins in quantitative microscopy, e.g., for correctly determining protein densities, interaction, clustering, and dynamics, requires a thorough characterization of their photophysical properties. One of the most important and most difficult to measure of these properties is the fluorescence quantum yield (QY), which characterizes the intrinsic brightness, i.e., how efficiently an FP converts absorbed light into fluorescence.

The most commonly used methods of QY measurement are the integrating sphere or comparison to a reference sample[5]. Both approaches are based on absolute brightness measurements and require knowledge about the absorption coefficient of the fluorescent molecules. However, absorption measurements become inaccurate if it is not precisely known how many of the light-absorbing molecules are indeed fluorescent and how many are in a non-fluorescent state (non-matured proteins, proteins in photophysical dark states). This can lead to gross underestimations of the QY, because non-luminescent fractions do not participate in fluorescence generation but still count in the QY determination based on measurements of absolute emission intensity. In contrast, a QY measurement that relies solely on measuring emission properties such as the fluorescence lifetime will be completely insensitive to the potential presence of non-radiative fractions. This is particularly important for FPs, which usually exhibit a plethora of non-fluorescent states due to specific structural conformations, protonation, chemical modifications, or incomplete maturation[2,6,7]. This may explain some of the large discrepancies of QY values reported in the literature (see Tables 1 and 2).

Recent years have witnessed the development of alternative ways to measure QY that are based on the ratio of the radiative ($k_r$) to the non-radiative ($k_{nr}$) de-excitation rate of a fluorophore[8]. Modulation of either the radiative or the non-radiative rate can be monitored by measuring the total rate ($k_r + k_{nr}$), i.e., the inverse

excited-state lifetime. Whereas the non-radiative rate of a fluorophore is typically determined by its intrinsic properties and local chemical environment, the probability of radiative de-excitation can be tuned by changing the so-called local density of states of the electromagnetic field[9]. Controlled tuning of the de-excitation rate of fluorophores has been achieved by placing them close to a dielectric interface[10,11], a sharp tip of a scanning probe microscope[12], a metallic mirror[13,14], a metallic nanoparticle[15,16], or between two gold nanoparticles[17]. However, all the above methods can be used for measuring fluorophores bound to a surface either physically or chemically, and hence are not applicable for FPs in their natural condition in buffer. An alternative way that allows one to modulate the radiative rate of fluorophores while keeping them freely floating in buffer is placing a droplet of solution between two metallic mirrors and tuning the distance between them. We have used this, so-called plasmonic nanocavity-based method, for measuring QY of dye molecules[18–21] and semiconductor nanocrystals[22,23].

Here we use the plasmonic nanocavity method[19,20] for absolute measurements of the QY of a broad class of FPs. The method is based on the modulation of the excited-state lifetime of a fluorophore by its interaction with a reflecting nanocavity that has a mirror distance in the range of one half the wavelength of light ($\lambda/2$ region), where the excited-state lifetime modulation is maximal[24]. The cavity modulates only the radiative transition of an FP from its excited to its ground state, but does not change its non-radiative transition, which is usually caused by the sterical hindrance of specific cis–trans isomerizations in the excited state and by excited-state proton transfer[25–27]. By measuring the total emission rate (inverse fluorescence lifetime) as a function of cavity size and then evaluating the measurements by using an exact semi-classical quantum-optical model, one can deduce absolute values of the QY of the fluorophore[19]. Thus, the nanocavity-based method does not require any preliminary calibration or comparison with a known sample. As the method is based exclusively on the measurement of the excited-state lifetime modulation, it is not prone to underestimating the QY due to light absorption by non-radiative species. The method can be used for any type of electric dipole emitter and was proven to be a reliable tool for determining QYs of dye molecules[19], semiconductor nanocrystals[23], or carbon nanodots[28]. It also allowed us to measure QYs of emitters in highly complex systems, such as mixtures of different types of fluorophores in liquid and solid

**Table 1 Values of the fluorescence quantum yield and excited-state lifetime of measured nonswitchable proteins in comparison with the literature values.**

| Protein | $\Phi_{meas}$ | $\Phi_{lit}$ | $\tau_{meas}$ (ns) | $\tau_{lit}$ (ns) |
|---|---|---|---|---|
| Citrine | 0.61 ± 0.01 | 0.54[34], 0.76[35] | 3.3 | 3.6[36] |
| EGFP | 0.61 ± 0.01 | 0.60[29,30] | 2.6 | 2.4–2.7[29] |
| mCherry | 0.24 ± 0.02 | 0.22[37–41], 0.23[42], 0.34[16] | 1.6 | 1.4[37], 1.5[33,42], 1.8[16] |
| Clover | 0.79 ± 0.01 | 0.76[40] | 3.1 | 3.2[43] |
| mEGFP | 0.59 ± 0.01 | 0.60[43,44], 0.73[45] | 2.6 | 2.59 [45] |
| mKate2 | 0.42 ± 0.01 | 0.39[42], 0.40[46], 0.60[16] | 2.5 | 2.5[42], 2.6[16] |
| mKO2 | 0.65 ± 0.01 | 0.62[47] | 3.5 | --- |
| mNeonGreen | 0.76 ± 0.01 | 0.80[43] | 3.0 | 3.1[43] |
| mOrange2 | 0.56 ± 0.01 | 0.60[46,48] | 2.5 | 2.7[48] |
| mPapaya | 0.80 ± 0.01 | 0.83[49] | 2.9 | --- |
| mPlum | 0.12 ± 0.03 | 0.10[46] | 1.0 | --- |
| mRuby2 | 0.35 ± 0.02 | 0.38[40], 0.45[42], 0.67[16] | 2.4 | 2.4[16], 2.5[42] |
| mTurquoise2 | 0.89 ± 0.01 | 0.93[50] | 3.8 | 4.0[50] |
| TagRFP | 0.51 ± 0.01 | 0.48[37] | 2.4 | 2.2–2.3[37] |
| TagRFP-T | 0.45 ± 0.01 | 0.41[46], 0.48[42] | 2.2 | 2.3[42] |

The error in all excited-state lifetime measurements did not exceed 0.1 ns.

**Table 2 Values of the fluorescence quantum yield and excited-state lifetime of measured photoswitchable proteins in comparison with the literature values.**

| Protein | $\Phi_{meas}^{100\%}$ | $\Phi_{meas}^{50\%}$ | $\Phi_{lit}$ | $\tau_{meas}^{100\%}(ns)$ | $\tau_{meas}^{50\%}(ns)$ | $\tau_{meas}^{10\%}(ns)$ | $\tau_{lit}(ns)$ |
|---|---|---|---|---|---|---|---|
| Dreiklang | 0.47 ± 0.01 | 0.45 ± 0.01 | 0.41[34] | 2.9 | 2.9 | 2.9 | --- |
| Dronpa | 0.69 ± 0.01 | 0.71 ± 0.01 | 0.68[31], 0.85[32] | 3.3 | 3.3 | 3.3 | --- |
| DronpaM159T | 0.22 ± 0.02 | 0.21 ± 0.01 | 0.23[51] | 1.2 | 1.2 | 1.2 | 0.6–0.9[52] |
| rsCherry | 0.22 ± 0.02 | 0.20 ± 0.01 | 0.02[53], 0.009[53] | 1.2 | 1.2 | 1.2 | --- |
| rsCherryRev | 0.08 ± 0.03 | 0.10 ± 0.01 | 0.005[41,53], 0.002[41], 0.0003[53] | 0.7 | 0.7 | 0.8 | --- |
| rsEGFP | 0.40 ± 0.01 | 0.40 ± 0.01 | 0.36[54] | 1.8 | 1.8 | 1.8 | --- |
| rsEGFP2 | 0.35 ± 0.02 | 0.34 ± 0.01 | 0.30[55] | 2.0 | 2.0 | 2.0 | --- |
| rsFastLime | 0.62 ± 0.01 | 0.59 ± 0.01 | 0.60[31], 0.77[51] | 3.1 | 3.1 | 3.1 | --- |

The error in all excited-state lifetime measurements did not exceed 0.1 ns.

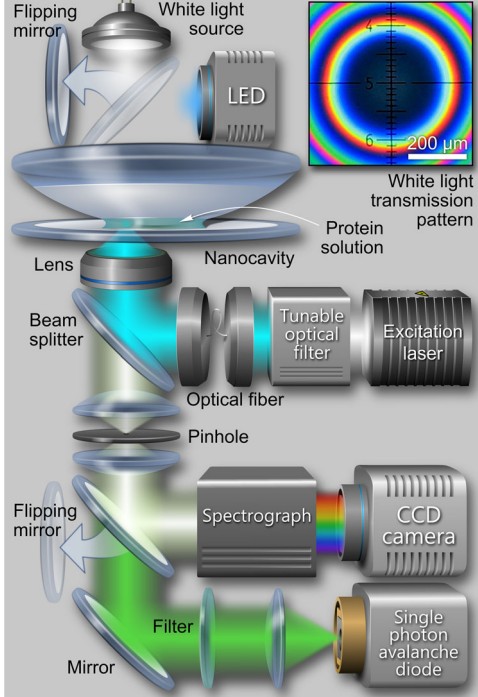

**Fig. 1 Schematic of the confocal scanning microscope and the plasmonic nanocavity that were used for measuring absolute quantum yields of proteins.** The inset shows the white-light transmission pattern around the center of the nanocavity. The first color ring corresponds to the λ/2 region of the cavity.

states with overlapping absorption and fluorescence spectra[22,23]. The high sensitivity of the technique allows one even to measure QYs of molecules inside single supported lipid bilayers[21], or even of individual fluorophores[18].

## Results
**Plasmonic nanocavity-based QY measurements.** Figure 1 shows a schematic of the custom-built confocal microscope and the nanocavity. The plasmonic nanocavity consists of two silver mirrors with subwavelength spacing. The bottom 30 nm- and top 60 nm-thick mirrors were prepared by vapor deposition of silver onto the surface of a commercially available and cleaned glass cover slide and a plano-convex lens (focal length of 150 mm), respectively. The spherical shape of the upper mirror allows for reversibly tuning the cavity size by moving the focal spot of the excitation laser laterally with a piezo nano-positioning stage.

Because of the very small curvature of the lens, the nanocavity can be considered to be plane-parallel across the focal spot even if it is located off the cavity's symmetry axis. The inset in Fig. 1 shows a white-light transmission image of the center of the nanocavity. The first color ring around the dark spot corresponds to the λ/2 region of the cavity, where the excited-state lifetime modulation of a fluorophore is maximized and where all QY measurements were done.

**Verification of the nanocavity-based method.** First, to verify the nanocavity-based method for measurement of FPs, we started with enhanced green fluorescent protein (EGFP), which has been well-studied using other methods[29,30]. Blue open circles in Fig. 2a show the results of the cavity-modulated excited-state lifetime measurements for EGFP in Tris buffer (at pH 7.5) at sub-micromolar concentration as a function of the cavity maximum transmission wavelength (linearly proportional to the distance between the mirrors). The error in all excited-state lifetime measurements did not exceed 0.1 ns. The solid blue curve is the theoretical fit to the experimental data that yields a QY value of 61 ± 1%. The obtained value is in excellent agreement with literature values that were obtained using standard methods[29,30]. The theoretical model that is used for data analysis takes into account the cavity geometry and the refractive index of the buffer solution. It yields an exact description of the cavity-induced modulation of the FPs' radiative transition rate (see ref. [19,20] for technical details). A key parameter when comparing theory and experiment is the distance between the cavity mirrors, which has to be known with nanometer accuracy. It was determined by measuring white-light transmission spectra and by fitting them with a standard Fresnel model of transmission through a stack of plano-parallel layers. Besides the QY value, the second free parameter of the theoretical model is the free-space (that is, out of cavity) excited-state lifetime of the fluorophore. We use it to verify the obtained QY values by comparing the free-space lifetime as obtained from the nanocavity measurement with that measured directly in a droplet of buffer on the surface of a glass cover slide. The discrepancy between the calculated and measured lifetime values does not exceed 5%, verifying the reliability of the obtained QY values. Thus, the nanocavity-based method allows one to precisely control all the parameters of the sample and to verify the obtained QY by the lifetime value, which is measured in a separate sample.

**QY of non-photoswitchable FPs.** In this study, we selected the commonly used FPs for deducing their absolute QY. Table 1 shows all QY values for non-photoswitchable proteins measured with the plasmonic nanocavity under identical experimental conditions as described above. The corresponding measured curves are shown in Supplementary Figs. 1–23. The shown

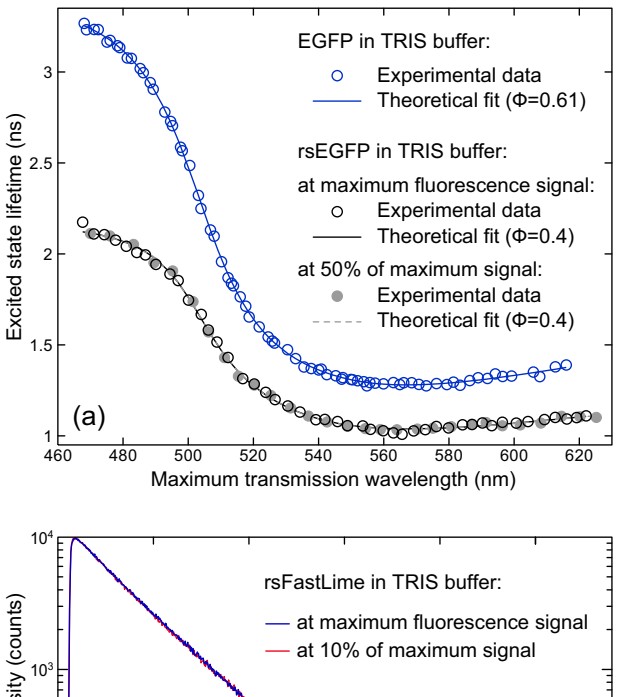

**Fig. 2 Excited-state lifetime measurements of fluorescent proteins. a** Excited-state lifetime of rsEGFP as a function of the maximum transmission wavelength (linearly proportional to the cavity length) of the nanocavity. Solid and open circles are data that were measured at maximum fluorescence signal and at 50% of maximum signal, respectively. The fit parameters are the fluorescence quantum yield (Φ) and the free lifetime in the absence of the cavity (τ). See Tables 1 and 2 for more details. **b** Fluorescence decay curves of rsFastLime at maximum fluorescence signal (blue curve) and at its 10% (red curve).

excited-state lifetime values have been measured separately in a droplet of Tris buffer on the surface of a clean glass cover slide, i.e., outside the nanocavity. Both the measured QYs and lifetimes are in very good agreement with the values published by other researchers for EGFP, mKate2, mKO2, mPapaya, and mPlum. The difference of 1–2% for QY and of 0.1 ns for lifetime can be attributed to measurement uncertainties. For some of the proteins the differences in QY values are considerably higher, which is likely to be caused by other factors. Moreover, the measured excited-state lifetimes and the literature values (if existent) also show significant differences. Interestingly, lower literature QY values mainly correspond to lower values of the excited-state lifetime. This correlation can be explained by the relation between these terms: for a given fluorophore, both the QY

$$\Phi = k_r(k_r + k_{nr})^{-1} \qquad (1)$$

and the lifetime

$$\tau = (k_r + k_{nr})^{-1} \qquad (2)$$

are determined by the radiative ($k_r$) and non-radiative ($k_{nr}$) de-excitation rates of a fluorophore. Both of them decrease with increase of the non-radiative rate $k_{nr}$. Therefore, when the non-radiative rate changes due to modifications of the local chemical environment or because of changes of the fluorophore's conformation, QY and excited-state lifetime do either both increase or decrease. Similarly, modulation of the radiative rate, e.g., by a change of the solvent's refractive index does also lead to synchronous changes of QY and lifetime. Therefore, the correlation of differences between measured and literature values of QY and lifetime suggests that these differences are not due to measurement error, but are inherent to the samples themselves. Probably, they are caused by differences in solvent properties, presence of dark states or incomplete purification of the sample that can lead to different fractions of fully maturated proteins. Because our nanocavity-based method is insensitive to the presence of non-luminescent dark states, one expects that our QY values reported here are either equal or larger than the values reported in the literature, provided that all other sample properties and conditions are the same.

**QY of photoswitchable proteins**. The high complexity of photoswitchable proteins makes measurements of their QY even more challenging and prone to errors, which is clearly visible from the discrepancy of literature values such as for Dronpa, rsFastLime and rsCherrryRev (see Table 2). Co-existence of proteins in on- and off-states, and overlap of their absorption spectra lead to systematic underestimation of QY when using conventional methods that are based on light emission and absorption measurements. As the nanocavity-based method does not require any absorption measurements, it is free from this drawback. For measurements of photoswitchable proteins, we additionally equipped the microscope with light emitting diodes (LEDs; see Fig. 1). This allowed us for switching proteins into the on-state in the nanocavity directly during the QY measurements by irradiation at a corresponding wavelength until fluorescence reached a maximum. Most of the nanocavity-determined QY values of photoswitchable proteins are in very good agreement with literature values. However, several measured values are considerably higher than the literature ones. We attribute this difference to the insensitivity of the nanocavity-based method to the presence of non-luminescent species that still absorb light but do not contribute to emission.

In case of rsCherry and rsCherryRev, the QYs determined by the nanocavity method differ by an order of magnitude from previously published values. We attribute this to the fact that in case of rsCherry and rsCherryRev the respective on- and off-states have similar absorption spectra and the proteins have short-lived on-states. Thereby, it is challenging to determine the fraction of on-state RSFPs in a solution, which makes conventional QY determinations prone to errors. As the nanocavity method is insensitive to the non-fluorescent species, this method provides a reliable measure of the QY of the on-state proteins.

Although the QY of Dronpa is in a good agreement with the value reported in ref. [31], it is significantly different from the value reported in ref. [32]. The absence of literature values for the excited-state lifetime for photoswitchable FPs do not allow us to unambiguously decide whether the observed difference in QY is due to differences in sample properties or due to measurement errors. However, the absolute manner of the nanocavity-based measurement ensures that in the current conditions the obtained results are correct.

**Residual fluorescence of protein ensembles in the dark state**. A key parameter of photoswitchable FPs, in particular for their use

in super-resolution microscopy, is the achievable contrast between bright and dark state. For the measurement of a given FP in the ensemble, it can be unclear whether the residual fluorescence of the dark state is the result of a low fluorescence of this state or the result of a small population of protein residing in the on-state. As the nanocavity-based method is not sensitive to the presence of non-radiative species within the sample, it allowed us to precisely measure the proteins' QY when they were switched into the partial off-state. Proteins were first switched from the brightest attainable fluorescent state into the partial off-state, where samples exhibited approximately half of their initial brightness. The measured cavity-induced modulation of the excited-state lifetime of rsEGFP at its maximum and 50% brightness as shown in Fig. 2a lead to nearly indistinguishable fits. Identity of both the QY and the excited-state lifetime values for the highest attainable and 50% brightness (Table 2) for all the proteins suggests that the transition from the on- to the off-state is a discrete one-step switching into a completely dark off-state. The proteins that remain in the on-state keep their QY unchanged.

To verify this result at extremely low fluorescence intensities, where nearly all the proteins are in the off-state, we measured excited-state lifetimes at <10% of maximum brightness observed. While QY measurement errors drastically grow at low values of sample brightness, the proportionality between the QY and the excited-state lifetime

$$\Phi = k_r \tau \tag{3}$$

(see Eqs. (1) and (2)) allows us to use the lifetime value as a proxy for the QY. Figure 2b shows fluorescence decay curves of rsFastLime measured at 100% and less than 10% of its brightness until the maximum of the curve reached $10^4$ counts. The curves are nearly identical, except the difference in background level, which is due to the difference in acquisition time. Identity of lifetime values obtained at 100%, 50%, and 10% sample brightness (see Table 2) confirms that the residual fluorescence originates from proteins in the on-state. This result shows the ability of the nanocavity-based method to discern QY values at low signal in a sample that contains various types of luminescent and non-luminescent species. This makes it advantageous over conventional methods based on light absorption measurement that leads to underestimation of QY in samples where non-emissive species are present. This advantage can be crucial in samples with incomplete purification or with fluorophores that can have different quantum states.

The lack of systematic excited-state lifetime measurements of photoswitchable FPs does not allow us to do any comparison of our measured and reported literature values. The striking difference of our and literature values of the lifetime of DronpaM159T could be caused either by intrinsic differences of the protein itself or by differences in temporal resolution of the measurement setup. The latter can lead to considerable differences between measurements, especially for short lifetime values, when the width of the instrument response function becomes comparable to the fluorescence decay time.

## Discussion

Using an absolute and calibration-free nanocavity-based method we have presented a comprehensive QY study of FPs that are widely used in microscopy. Some differences between our values and those reported in the literature demonstrate the difficulties inherent to conventional QY determination methods based on absolute intensity measurements, in particular for FPs with non-emissive fractions (non-matured proteins, presence of dark states). This is also the reason why our nanocavity-based method systematically yields larger QY values: only molecules in their fluorescent state contribute to our measurement, so that it is unbiased by the presence of non-luminescent species. The insensitivity of the nanocavity-based method to the presence of non-luminescent species allowed us to determine that for the analyzed proteins, the residual fluorescence of photoswitchable FPs in the ensemble originates from proteins residing in the on-state. A further systematic comparison of QY measurements using different methods will help to better understand the highly complex photo-physics of dark states in FPs that was observed in previous publications[16,33]. As the method does not need any calibration or a reference sample, it can be easily used not only in the visible spectral range, but also in the near infrared and ultraviolet ranges, where QY measurements are particularly challenging.

## Methods

**Expression and purification of FPs**. The proteins were expressed in the *Escherichia coli* strain Top 10 (Thermo Fisher Scientific, Waltham, MA, USA) using the pBAD/HisB or in the *E. coli* strain SURE (Agilent, Santa Clara, USA) using the pQE31 expression plasmid. *E. coli* cells were grown at 37 °C in lysogeny broth (LB) medium and protein expression was induced with 0.02% L-Arabinose or 1 mM isopropyl β-D-1-thiogalactopyranoside (IPTG).

The FPs were purified by Ni-NTA affinity chromatography (His SpinTrap Kit, GE Healthcare, Little Chalfont, BKM, GB) according to the manufacturer's instructions with a 30 min binding step. The purified proteins were concentrated by ultrafiltration using Vivaspin 500 columns (Sartorius, Göttingen, DE) and taken up in 100 mM Tris-HCl, 150 mM NaCl pH 7.5.

**Nanocavity preparation**. The cavity mirrors were prepared by vapor deposition of silver on the surface of a clean glass cover slide (bottom mirror) and a plane-convex lens (top mirror) by using a Laybold Univex 350 evaporation machine under high-vacuum conditions (~$10^{-6}$ mbar). The bottom and top mirrors had a thickness of 30 and 60 nm, respectively.

**Confocal microscope**. The distance between the cavity mirrors was monitored by measuring a white-light transmission spectrum using an Andor SR 303i spectrograph and a charge-coupled device camera (Andor iXon DU897 BV). By fitting these spectra with a standard Fresnel model of transmission through a stack of plan-parallel layers, one can determine the precise cavity length (distance between mirrors). Fluorescence lifetime measurements were performed with a custom-built confocal microscope equipped with an objective lens of high numerical aperture (Apo N, ×60/1.49 NA oil immersion, Olympus). A white-light laser system (Fianium SC400–4–20) with a tunable filter (AOTFnC-400.650-TN) served as excitation source. The excitation power was in the range below 10 μW at the sample plane. Collected fluorescence was focused onto the active area of a single photon detection module (MPD series, PDM). Data acquisition was accomplished with a multichannel picosecond event timer (PicoQuant HydraHarp 400). Photoswitching of proteins was performed using the light from LEDs Light Engine, Lumencore, Inc. The emission spectra of the diodes are shown in the Supplementary Information, Supplementary Fig. 24. The following excitation wavelengths were used:

440 nm: mTurquise2.

485 nm: Citrine, EGFP, Clover, mEGFP, mNeonGreen, Dreiklang, Dronpa, DronpaM159T, rsEGFP, rsEGFP2, rsFastLime.

509 nm: mKO2, mOrange2, mPapaya, mRuby2, TagRFP, TagRFP-T.

561 nm: rsCherry, rsCherryRev, mCherry, mKate2, mPlum.

**Fluorescence decay fitting**. Photon arrival times were histogrammed (bin width of 32 ps) for obtaining fluorescence decay curves. The obtained fluorescence decay curves were fitted with a multi-exponential decay model, from which the average excited-state lifetime was calculated according to

$$\langle \tau \rangle = \int_0^\infty F(t)t\,dt \Big/ \int_0^\infty F(t)\,dt$$

The multi-exponential decay model uses up to hundred different mono-exponential components that allows for fitting exponential decay of any complexity. Starting from a single mono-exponential function, the model stepwise increases the number of components used until further addition of components does not lead to improvement of the residual.

**Statistics and reproducibility**. For each protein type, data from at least two independently prepared samples was measured on separate days.

**Reporting summary**. Further information on research design is available in the Nature Research Reporting Summary linked to this article.

## Data availability

The datasets generated and analyzed during the current study are available in the Figshare repository via the link https://doi.org/10.6084/m9.figshare.12601205.v1 (see ref. [56]).

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

## Acknowledgements

This work was supported by the Deutsche Forschungsgemeinschaft (DFG, German Research Foundation) under Germany's Excellence Strategy—EXC2067/1–390729940. Financial support by German Research Foundation (DFG, SFB 937, project A14) is gratefully acknowledged.

## Author contributions

A.I.C., M.A., J.E., and S.J. designed the study. A.I.C. and M.A. performed the measurements. D.R. and A.I.C. analyzed the data. J.E., D.R., I.G., and A.I.C. developed the software for the data analysis. A.I.C., N.J., J.E., and S.J. wrote the manuscript.

## Funding

## Competing interests

The authors declare no competing interests.
