## [Peer Review File · Communications Biology]

Reviewers' comments:

Reviewer #1 (Remarks to the Author):

Manuscript: Absolute quantum yield measurements of fluorescent proteins using a plasmonic nanocavity

By Ruhlandt et al.

CommsBio

The authors report fluorescence quantum yields determined by using a plasmonic cavity that changes the radiative decay rate for a number of fluorescent proteins. Their finding largely agree with previously reported values for the quantum yield using conventional methods.

It is well known that fluorescent proteins are photophysically complex, which includes the presence of various emitting states as well as non-emitting states. There is a large number of publications that report on the presence of dark states (e.g. Hendrix et al. Biophys. J. 2008) and these dark states are known to complicate the determination of fluorescence quantum yield. Hence, determining the quantum yield using a method that is insensitive to dark states is potentially interesting.

Below I restrict myself and address some of the main points:

- It is highly surprising that in the presented study the authors overall find no difference to methods that include dark states. Numerous studies found evidences for dark states, yet the authors find no sign of these states. Unfortunately this is not discussed by the authors
- A recent study, cited in the manuscript (Prangma, Physical Chemistry B, 2020) did not only evidenced the presence of dark states, but also quantified the fraction of dark states – including fluorescent proteins measured by the authors. This paper contains data clearly in contradiction to the results presented. This is not mentioned, nor discussed.
- The data presented is extremely limited. These omissions make it hard/impossible to realistically evaluate the paper.
 - o There is no data showing the observed decays in the cavity.
 - o The information given on the data analysis is very rudimentary. According to the experimental methods, decays were fitted to a multiexponential decay model. How many decays? Why this number?
 - o How were multiple emitting states that often give rise to non-single exponential decay in solution (even in absence of cavity) addressed in the analysis of the decays in the cavity?
- The few data the authors show is the change of observed lifetime at different cavity distances for EGFP. The quantum yield obtained for EGFP agrees with conventional measurements that did not exclude dark states. The authors use this to evidence the reliability of their measurement. Note: One of the two papers the authors cite in this context only marginally addresses EGFP and even states “the quantum yield of EGFP was assumed to be 0.60”. Moreover the quantum yield of the bright state, excluding dark states, just as the authors report, has been published before for EGFP and considerably deviates from what the authors report (once more see Prangma et al, citation therein).
- Whenever the authors find noticeable deviations between their and (some) of the values reported previously based on measurements that include dark states, the values reported here are on the lower side. This is remarkable, given that they claim that their method excludes dark states, hence I would expect their values to be higher than the previously reported values

There are more points that could be mentioned. As it is the paper does not offer new insights. The data presented suggests that the emitting state quantum yield is identical to the quantum yield when bright and dark states are averaged. Such finding seems little plausible and is also in opposition to earlier findings. Overall, to my opinion the presented manuscript is not suitable for publication.

Reviewer #2 (Remarks to the Author):

The manuscript by D. Ruhlandt et al. deals with the quantum yield determination of different fluorescent proteins. The method employed has been previously introduced almost a decade ago by some of the authors (references 1 and 2 of the current manuscript). Nevertheless, I think the manuscript has sufficient elements of novelty to be considered for publication in Communications Biology. The authors provide an impressive table with an accurate determination of lifetime and quantum yield for several fluorescent proteins which will help researchers in the field to design and interpret the results of their experiments. Therefore, I would be in favor of publication after the following minor comments are addressed:

- On page 4, the authors write: "The cavity modulates only the radiative transition of a fluorophore from its excited to its ground state, but does not change its non-radiative transition, which is usually caused by collisional interactions of a fluorophore with surrounding molecules" I think instead of from the authors mean from
- On page 4, the authors write: "The controlled tuning of the radiative rate has been shown for fluorophores placed close to a dielectric interface^{12, 13}, a sharp tip of a scanning probe microscope¹⁴, a metallic mirror^{15, 16}, a metallic nanoparticle^{17, 18}, or between two gold nanoparticles¹⁹." I think the authors could be more precise. The presence of for example metallic nanoparticles would also affect the non-radiative rates and lead to quenching. This might be understood as an increment of the overall non-radiative decay, or as an extra non-radiative rate to be added to the "intrinsic" non-radiative rate, nevertheless and regardless of the interpretation, this will lead to changes in the measured intensity and lifetime. Based on the line by authors the reader might get the impression that only the radiative rate is affected.
- I found no separation of figure 2 into a and b. Also, I found no caption for what I guess is part b.
- The authors include no error for the lifetime measurements. Also, I would appreciate more discussion about the difference in the measured and literature value of the excited lifetime of DronpaM159T. Such measurements should be easy to perform.
- I think the authors should provide more raw data (perhaps in the supporting information)

Reviewer #3 (Remarks to the Author):

The Manuscript by Ruhlandt et al. reports on the validation of plasmonic nanocavity-based approach for quantum yield measurement of famous biotechnology reporters – genetically encoded fluorescent proteins. The Authors highlight the advantages of this calibration-free method and low requirements for the sample size. The introduction of an independent QY measurement approach holds the promise of solving many discrepancies observed previously between single-molecule brightness of fluorescent proteins, the spectrophotometry data obtained from bulk samples of protein solutions, and the results of quantum chemistry predictions.

The suggested method is not a replacement for conventional approaches but is an important complementary one. The Referee welcomes the addition of the nanocavity QY estimation method to the arsenal of the methods for FP characterization and assesses the data already included in the paper as very interesting for FP-related scientific community. The Referee would like to see, in the future, the nanocavity QY data for most published FPs. Regardless of interpretation, the Referee believes that the provided experimental data merit the publication.

The Referee would like to share the following comments on the Manuscript.

- 1) The authors should be more cautious in stating that (P4) "the cavity modulates only the radiative transition" and summing all non-radiative transitions to "collisional interactions of a fluorophore with surrounding molecules." The Referee believes that the sterical restriction of certain cis-trans isomerization in the excited state and excited-state proton transfer are considered

as major factors determining QY of fluorescent proteins (i.e., doi: 10.1126/science.1207339, doi: 10.1021/ja3010144, doi: 10.1021/acs.jctc.5b00894).

2) (Following the same line of concern). As stated in Ref.1, "A necessary prerequisite for successfully using the cavity-modulated lifetime for accurate QY measurements is an accurate theoretical understanding of the underlying physics." For the sake of the general reader, explicitly state whether the assumptions included in the model in Ref.1 hold, to the best of the authors' understanding, for fluorescent proteins.

3) In some published works, the FPs are reported as quite unconventional fluorophores, where the chromophore is subjected to strong intrinsic electric fields, essentially determining their spectral properties (i.e., doi: 10.1021/jp907085p). Please comment whether the existence of such strong fields may hinder the cavity-induced modulation of the radiative transition rate of fluorescent proteins or affect the calculations.

4) The Referee did not get the 'differences of dimerization' part in the discussion of discrepancies between measured and literature QY (P6). Please, elaborate.

5) Please check whether full details on the measurement are included in the corresponding sections of the Manuscript. For instance, what were the exact wavelengths used for excitation for each protein tested? Please also include the data on the laser powers used, preferably measured at the sample plane.

Details on the LED used for photoconversion are not included in the text.

Please be slightly more precise in reference to 'extremely low fluorescence intensities' (P8)

6) The Referee invites the Authors to provide raw data behind tables 1 and 2 on FigShare (preferably, in the form allowing for independent fitting, something like the source for top panel in Figure 2), as this data is clearly of considerable interest for computational chemistry.

Typo P4 "radiative transition of a fluorophore form" -> from

Dear Sir/Madam,

We are grateful for your positive evaluation of the manuscript and for your comments that allowed us to significantly improve the quality of the manuscript. Below are the detailed replies to your comments.

Sincerely yours,

Alexey Chizhik

Reviewer #1 (Remarks to the Author):

Comment: The authors report fluorescence quantum yields determined by using a plasmonic cavity that changes the radiative decay rate for a number of fluorescent proteins. Their finding largely agree with previously reported values for the quantum yield using conventional methods. It is well known that fluorescent proteins are photophysically complex, which includes the presence of various emitting states as well as non-emitting states. There is a large number of publications that report on the presence of dark states (e.g. Hendrix et al. Biophys. J. 2008) and these dark states are known to complicate the determination of fluorescence quantum yield. Hence, determining the quantum yield using a method that is insensitive to dark states is potentially interesting.

Reply: We thank the author for pointing out that a more detailed discussion of the influence of dark states on quantum yield measurements is required. We also agree with the reviewer that the publication by Hendrix et al. contains relevant measurement results. Although the article by Hendrix et al. does not report about quantum yield measurements, their values of excited state lifetimes are nearly identical to those measured in our work. This shows that the samples used in their study possess photo-physical properties that are very similar to ours. Therefore, we assume that also the quantum yields of their samples will be very close to our values. We have added the following sentence to extend the discussion of this point in the manuscript:

“A further systematic comparison of QY measurements using different methods will help to better understand the highly complex photo-physics of dark states in FPs that was observed in previous publications.”

Also, according to the request of the reviewer, we added the following reference:

Hendrix, J., Flors, C., Dedecker, P., Hofkens, J. & Engelborghs, Y. Dark States in Monomeric Red Fluorescent Proteins Studied by Fluorescence Correlation and Single Molecule Spectroscopy. *Biophysical Journal* **94**, 4103-4113 (2008).

Comment: It is highly surprising that in the presented study the authors overall find no difference to methods that include dark states. Numerous studies found evidences for dark states, yet the authors find no sign of these states. Unfortunately this is not discussed by the authors.

Reply: We agree with the reviewer that previous publications that highlight the presence of dark states in fluorescence proteins had not been adequately discussed in the original version of our manuscript. Now, we have added references to all the relevant publications as suggested by the reviewer and provided a comparison to their results in Table 1:

Hendrix, J., Flors, C., Dedecker, P., Hofkens, J. & Engelborghs, Y. Dark States in Monomeric Red Fluorescent Proteins Studied by Fluorescence Correlation and Single Molecule Spectroscopy. *Biophysical Journal* **94**, 4103-4113 (2008).

and

Prangma, J.C. et al. Quantitative Determination of Dark Chromophore Population Explains the Apparent Low Quantum Yield of Red Fluorescent Proteins. *The Journal of Physical Chemistry B* **124**, 1383-1391 (2020).

We provide now a comparative list of proteins that were measured in the above article. The results of measurements and literature values are included in Table 1.

Comment: A recent study, cited in the manuscript (Prangma, Physical Chemistry B, 2020) did not only evidenced the presence of dark states, but also quantified the fraction of dark states – including fluorescent proteins measured by the authors. This paper contains data clearly in contradiction to the results presented. This is not mentioned, nor discussed.

Reply: We thank the reviewer for pointing us to the work by Prangma et al.. A comparison of our results shown with those of this publication shows that whereas the quantum yield values show considerable differences, the excited state lifetime values agree very well. We attribute the discrepancy between the quantum yield values to the fact that Prangma et al. studied proteins embedded in a dry polymer film, which can significantly change their photophysical properties. This makes a direct quantitative comparison of protein properties questionable. We have measured our

values in buffer because this is the more relevant environment for almost all applications of FPs in biology and biophysics.

Nonetheless, for the sake of completeness, we added the values obtained by Prangma et al. to Table 1, and added the following sentence to the manuscript:

“A further systematic comparison of QY measurements using different methods will help to better understand the highly complex photo-physics of dark states in FPs that was observed in previous publications.”

Comment: There is no data showing the observed decays in the cavity.

Reply: We are grateful to the reviewer for pointing out that the data used for the calculations must be made available for readers. We uploaded the data and the routines used in this study at FigShare, in the project entitled “Absolute quantum yield measurements of fluorescent proteins using a plasmonic nanocavity”.

Comment: The information given on the data analysis is very rudimentary. According to the experimental methods, decays were fitted to a multiexponential decay model. How many decays? Why this number?

Reply: We agree with the reviewer that a more detailed explanation about decay curve fitting must be provided. We added a detailed description in the section “Fluorescence decay fitting” to the Methods.

Comment: How were multiple emitting states that often give rise to non-single exponential decay in solution (even in absence of cavity) addressed in the analysis of the decays in the cavity?

Reply: Indeed, the complex photophysics of FPs is often reflected by a non-trivial multiexponential decay behavior of their fluorescence decay curves. Therefore, fitting of all measured decay curves was done by a non-linear non-negative least-square fit of a distribution of exponential decay functions with logarithmically spaced decay times. Such an approach thus not require an a priori knowledge of the number of discrete decay components. By step-wise increasing the number of mono-exponential components used for fitting, the routine monitors whether addition of more component does indeed reduce the residuals between fit and experimental values. The fitting process ends when no further

change in the residuals is observed. A detailed explanation as well as a link to the routine is added to the section “Fluorescence decay fitting” in Experimental Methods.

Comment: The few data the authors show is the change of observed lifetime at different cavity distances for EGFP. The quantum yield obtained for EGFP agrees with conventional measurements that did not exclude dark states. The authors use this to evidence the reliability of their measurement. Note: One of the two papers the authors cite in this context only marginally addresses EGFP and even states “the quantum yield of EGFP was assumed to be 0.60”. Moreover the quantum yield of the bright state, excluding dark states, just as the authors report, has been published before for EGFP and considerably deviates from what the authors report (once more see Prangma et al, citation therein).

Reply: We agree with the reviewer that the presence of dark states can lead to an underestimation of quantum yield values when using methods that are based on measuring light absorption. The difference between our nanocavity-based results and some literature values (Tables 1 and 2) can indeed be due to the presence of dark states. However, the very good agreement of our and published values for EGFP and several other proteins suggests that dark states, even if present, play a rather negligible role for these proteins. As discussed in the original version of our manuscript, we do not exclude that differences in sample preparation can also result in differences in quantum yield values: “...the correlation of differences between measured and literature values of QY *and* lifetime suggests that these differences are not due to measurement errors, but are inherent to the samples themselves. Probably, they are caused by differences in solvent properties, presence of dark states or incomplete purification of the sample that can lead to different fractions of fully matured proteins.” However, following the reviewer’s request, we highlight that these differences can also be due to the presence of dark states, and we added a corresponding sentence to the text.

Comment: Whenever the authors find noticeable deviations between their and (some) of the values reported previously based on measurements that include dark states, the values reported here are on the lower side. This is remarkable, given that they claim that their method excludes dark states, hence I would expect their values to be higher than the previously reported values.

Reply: We agree with the reviewer that our insensitivity to non-luminescent species of dark states should lead to increased values of quantum yield. This is indeed the case for the following proteins: Clover, mKO2, mPlum, TagRFP, Dreiklang, rsCherry, rsCherryRev rsEGFP, rsEGFP2 and rsFastLime. For several other proteins, the literature values show a considerable difference and are both above or

below our values. For some of them, we obtain indeed QY values below the ones reported in the literature. We attribute this to differences in solvent properties or incomplete purification of the sample. To clarify this point, we added the following text to the manuscript:

“Probably, they are caused by differences in solvent properties, presence of dark states or incomplete purification of the sample that can lead to different fractions of fully matured proteins. Because our nanocavity-based method is insensitive to the presence of non-luminescent dark states, one expects that our QY values reported here are either equal or larger than the values reported in the literature, provided that all other sample properties and conditions are the same.”

Reviewer #2 (Remarks to the Author):

Comment: The manuscript by D. Ruhlandt et al. deals with the quantum yield determination of different fluorescent proteins. The method employed has been previously introduced almost a decade ago by some of the authors (references 1 and 2 of the current manuscript). Nevertheless, I think the manuscript has sufficient elements of novelty to be considered for publication in Communications Biology. The authors provide an impressive table with an accurate determination of lifetime and quantum yield for several fluorescent proteins which will help researchers in the field to design and interpret the results of their experiments. Therefore, I would be in favor of publication after the following minor comments are addressed.

Reply: We thank the reviewer for the positive evaluation of the manuscript.

Comment: On page 4, the authors write: “The cavity modulates only the radiative transition of a fluorophore from its excited to its ground state, but does not change its non-radiative transition, which is usually caused by collisional interactions of a fluorophore with surrounding molecules” I think instead of from the authors mean from

Reply: We are grateful to the reviewer for carefully reading the manuscript. We corrected this typo.

Comment: On page 4, the authors write: “The controlled tuning of the radiative rate has been shown for fluorophores placed close to a dielectric interface^{12, 13}, a sharp tip of a scanning probe microscope¹⁴, a metallic mirror^{15, 16}, a metallic nanoparticle^{17, 18}, or between two gold nanoparticles¹⁹.” I think the authors could be more precise. The presence of for example metallic nanoparticles would also affect the non-radiative rates and lead to quenching. This might be understood as an increment of the overall non-radiative decay, or as an extra non-radiative rate to be added to the “intrinsic” non-radiative rate, nevertheless and regardless of the interpretation, this will lead to changes in the measured intensity and lifetime. Based on the line by authors the reader might get the impression that only the radiative rate is affected.

Reply: We agree with the reviewer that some of the above cases can indeed involve a very complex modulation of both radiative and non-radiative rates. To prevent any confusion, we changed the sentence in the manuscript to:

“Controlled tuning of the de-excitation rate of fluorophores has been achieved by placing them close to a dielectric interface, a sharp tip of a scanning probe microscope, a metallic mirror, a metallic nanoparticle, or between two gold nanoparticles.”

Comment: I found no separation of figure 2 into a and b. Also, I found no caption for what I guess is part b.

Reply: We are grateful to the reviewer for pointing this out. We increased the font of the labels (a) and (b) in figure 2. The caption was amended.

Comment: The authors include no error for the lifetime measurements.

Reply: The errors of all lifetime measurements did not exceed 0.1 ns, as has been indicated in the captions of Tables 1 and 2. In the revised version of the manuscript, we also added the following sentence to the main text:

“The error in all excited state lifetime measurements did not exceed 0.1 ns.”

Comment: Also, I would appreciate more discussion about the difference in the measured and literature value of the excited lifetime of DronpaM159T. Such measurements should be easy to perform.

Reply: We thank the reviewer for pointing out that the striking difference between the measured and literature lifetime value of DronpaM159T is not properly discussed. We do not have a simple explanation for this fact, since it could only be obtained by a direct comparison of samples and microscopes identical to those used in these studies. However, we assume that a certain underestimation of the average lifetime in ref.52 may be due to the relatively broad instrument response function often observed when using pulsed diode lasers for excitation. Another possible contribution to its broadening could also be caused by the limited temporal resolution of the employed photo-detectors. To clarify this point, we added the following discussion to the manuscript: “The lack of systematic excited state lifetime measurements of photoswitchable FPs does not allow us to do any comparison of our measured and reported literature values. The striking difference between our and literature values of the lifetime of DronpaM159T could be caused either by intrinsic differences of the protein itself or by differences in temporal resolution of the measurement setups. The latter can lead to considerable differences between measurements, especially for short lifetime

values, when the width of the instrument response function becomes comparable to the fluorescence decay time.”

Comment: I think the authors should provide more raw data (perhaps in the supporting information).

Reply: Following the request of the reviewer, the data was uploaded to FigShare and was added to the Supplementary Information.

Reviewer #3 (Remarks to the Author):

Comment: The Manuscript by Ruhlandt et al. reports on the validation of plasmonic nanocavity-based approach for quantum yield measurement of famous biotechnology reporters – genetically encoded fluorescent proteins. The Authors highlight the advantages of this calibration-free method and low requirements for the sample size. The introduction of an independent QY measurement approach holds the promise of solving many discrepancies observed previously between single-molecule brightness of fluorescent proteins, the spectrophotometry data obtained from bulk samples of protein solutions, and the results of quantum chemistry predictions. The suggested method is not a replacement for conventional approaches but is an important complementary one. The Referee welcomes the addition of the nanocavity QY estimation method to the arsenal of the methods for FP characterization and assesses the data already included in the paper as very interesting for FP-related scientific community. The Referee would like to see, in the future, the nanocavity QY data for most published FPs. Regardless of interpretation, the Referee believes that the provided experimental data merit the publication.

Reply: We are grateful to the reviewer for the positive evaluation of our manuscript.

Comment: The authors should be more cautious in stating that (P4) "the cavity modulates only the radiative transition" and summing all non-radiative transitions to "collisional interactions of a fluorophore with surrounding molecules." The Referee believes that the sterical restriction of certain cis-trans isomerization in the excited state and excited-state proton transfer are considered as major factors determining QY of fluorescent proteins (i.e., doi: 10.1126/science.1207339, doi: 10.1021/ja3010144, doi: 10.1021/acs.jctc.5b00894).

Reply: We agree with the reviewer that the non-radiative de-excitation in fluorescent proteins could be also due to other mechanisms. To clarify this point, we modified the following sentence in the manuscript:

“The cavity modulates only the radiative transition of an FP from its excited to its ground state, but does not change its non-radiative transition, which is usually dominantly determined by cis-trans isomerizations in the excited state and by excited-state proton transfer.”

Additionally, we added the references suggested by the reviewer.

Comment: (Following the same line of concern). As stated in Ref.1, "A necessary prerequisite for successfully using the cavity-modulated lifetime for accurate QY measurements is an accurate theoretical understanding of the underlying physics." For the sake of the general reader, explicitly state whether the assumptions included in the model in Ref.1 hold, to the best of the authors' understanding, for fluorescent proteins.

Reply: We thank the reviewer for pointing out that Ref.1 does not explicitly clarify whether the model used in this work does also hold for fluorescent proteins. The model that is used for the nanocavity-based method is not limited to certain types of fluorophores, but can be used for quantum yield measurements of any electric dipole emitter. To clarify this point, we have added the following sentence to the manuscript:

"The method can be used for any type of electric dipole emitter and was proven to be a reliable tool for determining QYs of dye molecules, semiconductor nanocrystals, or carbon nanodots."

Comment: In some published works, the FPs are reported as quite unconventional fluorophores, where the chromophore is subjected to strong intrinsic electric fields, essentially determining their spectral properties (i.e., doi: 10.1021/jp907085p). Please comment whether the existence of such strong fields may hinder the cavity-induced modulation of the radiative transition rate of fluorescent proteins or affect the calculations.

Reply: To the best of our knowledge, the nanocavity-based method can be used for any type of emitter. The current version of the model that has been used in this study assumes that fluorescence stems from electric dipole emitters. There are no other assumptions or restrictions that limit its applicability. The presence of static intrinsic electric fields does not change the validity of our data analysis. The model can be adapted also to other more complex emitters, e.g. electric quadrupole or magnetic dipole emitters. However, this extension is irrelevant in the context of fluorescent proteins and, therefore, goes far beyond the scope of the present work.

Comment: The Referee did not get the 'differences of dimerization' part in the discussion of discrepancies between measured and literature QY (P6). Please, elaborate.

Reply: We thank the reviewer for pointing out that differences in dimerization that can indeed be

observed in rather specific cases are not relevant to the current work. We amended the corresponding sentence as follows:

“Probably, they are caused by differences in solvent properties, presence of dark states, or incomplete purification of the sample that can lead to different fractions of fully matured proteins.”

Comment: Please check whether full details on the measurement are included in the corresponding sections of the Manuscript. For instance, what were the exact wavelengths used for excitation for each protein tested? Please also include the data on the laser powers used, preferably measured at the sample plane. Details on the LED used for photoconversion are not included in the text. Please be slightly more precise in reference to 'extremely low fluorescence intensities' (P8)

Reply: We thank the reviewer for pointing out that more details of the experimental conditions should be provided. We added all the requested information to the Methods, section “Confocal microscope”.

Comment: The Referee invites the Authors to provide raw data behind tables 1 and 2 on FigShare (preferably, in the form allowing for independent fitting, something like the source for top panel in Figure 2), as this data is clearly of considerable interest for computational chemistry.

Reply: Following the reviewer’s request, we have uploaded the measured data for all proteins to FigShare in the project entitled “Absolute quantum yield measurements of fluorescent proteins using a plasmonic nanocavity”.

Comment: Typo P4 "radiative transition of a fluorophore form" -> from

Reply: We are grateful to the reviewer for careful reading of the manuscript. The typo is corrected.

REVIEWERS' COMMENTS:

Reviewer #3 (Remarks to the Author):

The Manuscript by Ruhlandt et al. has improved. The Authors amended the text, included clarifications and relevant citations. The Authors provided raw data and Matlab scripts behind the analysis as Figshare dataset.

The Referee endorses the Manuscript in its current form.

Do not forget to cite Figshare dataset with doi.